# Can LLMs Imitate Social Media Dialogue? Techniques for calibration and BERT-based Turing-Test

## Abstract

Large language models (LLMs) are increasingly used to simulate human behavior in online environments, yet existing evaluation methods, e.g., simplified Turing tests with human annotators, fall short of capturing the subtle stylistic and affective features that distinguish human- from AI-generated text. In this study, we introduce a human-likeness evaluation framework that systematically quantifies how closely LLM-generated social media replies resemble those written by real users. Our framework leverages a suite of interpretable textual features capturing stylistic, tonal, and emotional dimensions of online conversation. We apply this framework to evaluate five commonly used open-weight LLMs across a variety of generation configurations, including fine-tuning, stylistic few-shot prompting, and context retrieval. To benchmark and enhance realism, we incorporate a machine learning–based judge that ranks candidate AI responses according to their similarity to human replies. Our results reveal persistent divergences between human and LLM-generated replies, especially in affective and stylistic dimensions. Nonetheless, we identify clear gains in realism from stylistic conditioning, context-aware prompting, and fine-tuning, with models such as Gemma, Llama, and Mistral performing best.

## 1 Introduction

Large Language Models (LLMs) have rapidly become key tools in the social sciences, supporting tasks ranging from data annotation and synthetic data generation to survey design (Törnberg et al., 2023; Gilardi et al., 2023; Ziems et al., 2024). Increasingly, researchers are leveraging LLMs to simulate human behavior, drawing on their capacity to mimic conversational patterns and decision-making processes. This role-playing unlocks new possibilities: LLMs can serve as controllable, consistent, and scalable confederates in experiments with human participants (Argyle et al., 2023a; Flamino et al., 2024), or power new forms of social simulation that move beyond the constraints of conventional agent-based models (Park et al., 2023; Guo et al., 2024; Liu et al., 2025). By generating discourse and imitating human-like decisions while conditioning on demographic attributes or past conversations, LLMs promise to capture nuance that traditional approaches miss (Argyle et al., 2023b).

Social media has become a key application area for these methods (e.g., Törnberg et al. (2023)). A growing body of work uses generative simulations to study emergent phenomena on social networks or explore counterfactual scenarios (Orlando et al., 2025; De Marzo et al., 2023). However, for both experiments and simulations, the believability of LLM-generated dialogue is crucial. If the language fails to convincingly mimic human discourse, it can bias participant reactions or lead simulations to produce misleading results. For example, studies have found that, when using LLMs in experiments, participants perceive LLM confederates as less convincing than real humans (Flamino et al., 2024), and researchers have highlighted validation and calibration as core challenges for generative social simulation (Larooij & Törnberg, 2025; Bail, 2024; Anthis et al., 2025; Grossmann et al., 2023).

This raises an important question: how human-like is LLM-generated discourse, and how can we enhance its realism? Current evaluations typically rely on human judgments of "believability" — testing whether people can distinguish between human- and machine-

generated text (Park et al., 2023). Yet this approach has serious limitations (Larooij & Törnberg, 2025). Humans often fail to detect flaws, setting a low bar for human-likeness. Moreover, such tests overlook the subtle linguistic, emotional, and social cues that characterize authentic human communication. This is especially problematic for social science applications that seek to model phenomena like toxic behavior, polarization, or emotional contagion: it is not enough for LLM outputs to appear superficially human—they must faithfully reproduce the tone, structure, and affective texture of real social media discourse. Furthermore, on the calibration side, most research has relied on prompt engineering rather than more advanced fine-tuning techniques, suggesting room for significant improvement.

In this paper, we investigate the extent to which LLMs can generate replies that resemble real-world social media discourse. We focus on X/Twitter reply threads, where we compare human-authored responses to alternatives generated by different LLMs. Our contributions are threefold. First, we introduce a human-likeness evaluation framework that goes beyond traditional human-judgment tests: we finetune a BERT-based classifier to distinguish between human- and AI-generated replies. Second, we benchmark a range of open-weight LLMs under various configurations — including finetuning, few-shot prompting, and context retrieval — on their ability to produce replies that evade detection. We further propose a machine learning–based ranking mechanism (ML-judge) to select the most human-like output for each prompt, offering a scalable path toward enhancing stylistic realism. Third, we conduct a detailed feature analysis to identify the linguistic and psychological markers that differentiate human from machine-generated text, examining both style metrics and psychological attributes such as sentiment, toxicity, and emotional content.

Our analysis reveals persistent stylistic disparities between human- and AI-generated text. We find that while fine-tuning, stylistic examples, and context retrieval can reduce detectability, especially in models like Google-gemma-3-4B-Instruct, no configuration fully evades classification. Even when responses are optimized through ML-based selection, AI-generated text remains less toxic, more positive, and stylistically distinct from human-authored content. These results highlight that genuine human-likeness in generative text depends not only on model architecture but also on sophisticated output selection and finer control over stylistic markers. This analysis sheds light on the persistent gaps that limit current LLMs' ability to mimic authentic human behavior.

## 2 Related Work

A growing body of research explores how generative AI, particularly large language models (LLMs), can simulate human behavior for social science applications (Guo et al., 2024; Xi et al., 2023). This literature spans efforts to model individuals and groups, evaluate the fidelity of generated content, and use generative agents in multi-agent simulations. Our work builds on this foundation, with a focus on benchmarking the stylistic human-likeness of LLM-generated responses in social media environments.

LLMs have shown promise in mimicking human behavior when prompted with social or psychological context. Argyle et al. (2023b) demonstrated that LLMs can generate survey-style responses reflective of different demographic groups, sparking interest in using these models as stand-ins for human participants. Bail (2024), Ziems et al. (2024), and Davidson (2024) articulate broader arguments for the role of generative AI in advancing empirical research, while also noting the need for methodological safeguards.

LLMs have also been used to simulate populations of interacting agents. Park et al. (2023) introduced generative agents that emulate human-like memory, planning, and interaction. This line of work has expanded to large-scale simulations of civic life (Park et al., 2024), social networks (Gao et al., 2023), and online communities (Liu et al., 2025), where the realism of agent behavior is increasingly critical.

A compelling application of this paradigm is shown by Törnberg et al. (2023), who use LLM-driven personas to study the impact of news feed algorithms on political discourse. Using survey data from the American National Election Studies (ANES), they simulate a Twitter-like platform where agents interact under three feed designs: an echo chamber,

a global popularity feed, and a novel "bridging" algorithm. The bridging feed, which promotes cross-partisan engagement, leads to more ideologically diverse exposure and reduced toxicity. This work illustrates how generative agents can test interventions in complex social systems in a controlled, reproducible setting.

Despite their fluency, LLMs raise concerns about whether their outputs truly resemble human language – especially in informal, dynamic settings like social media. Several studies have assessed this "human-likeness," with Wang et al. (2024); Bisbee et al. (2024); Santurkar et al. (2023) warning that LLMs may flatten or misrepresent group-specific linguistic patterns, posing ethical and methodological risks. Others argue that LLMs' ability to produce convincing dialogue limits their use as confederates in human-subject experiments (Flamino et al., 2024). Thus the validity of such simulations depends crucially on the degree to which LLM-generated interactions resemble real human behavior, not just in content, but in linguistic style, tone, and sentiment.

Scholars have argued that the two central current challenges of generative social simulations is *validation* – how to show that the LLMs are reproducing realistic behavior – and *calibration* – how to align the LLMs with human behavior (Larooij & Törnberg, 2025). To contribute to these central aims for enabling realistic agent-based simulations, this paper provides a foundation for evaluating and improving the stylistic fidelity of LLM-generated responses, focusing on the case of social media dialogue. We focus on reproducing social media dialogue as it represents a relatively simple form of human dialogue, and hence provides a minimal competency task -— if the model fails here, its broader utility for mimicking realistic dialogue in social simulation is questionable.

## 3 Data & Methods

Our dataset builds on a set of social media users previously collected by Cerina (2025) and comprises Twitter/X conversations with tweet-reply pairs, tweet metadata, and user-level information. Each data point includes a tweet, its parent tweet, and the replying user's identity. We split the dataset into training and test sets, focusing our evaluation on 250 users with at least 20 replies in the test set (for each user, we randomly sampled 20 reply tweets).

Our goal is to simulate how each user might respond to a tweet using large language models (LLMs) and to evaluate the likelihood that AI-generated replies are stylistically similar to those of humans. To this end, we prompted LLMs to produce one-sentence responses, emulating each user's linguistic style and conversational behavior. We tested five families of open-weight LLMs, namely **DeepSeek**, **Gemma**, **Llama**, **Mistral**, and **Qwen**. More specifically we used: DeepSeek-R1-Distill-Llama 8B (DeepSeek-AI, 2025), Meta-Llama 3.1 8B (Meta Llama, 2024a), Mistral v-0.1 7B (Jiang et al., 2023), Google-Gemma 3 4B Instruct (Team, 2025), Meta-Llama 3.1 8B Instruct (Meta Llama, 2024b), Mistral v-0.1 7B Instruct (Jiang et al., 2023), Qwen 2.5 7B Instruct (Yang et al., 2024). Each model was used with temperature set to 0.8, and we tested four increasingly advanced configurations:

- **Baseline (BL)** configuration consisting of a simple prompt like:

  ```
  prompt = "[Instruction] You are @{username}. Continue the conversation naturally
       adding a concise (one sentence) tweet reply.\n"
   prompt+= "[Conversation] " + "\n".join(reply_to_message) + f"\n {username}:"
  ```

- **Stylistic Examples (SE)**: The prompt included 10 examples of the user's prior replies drawn from the training set.

- **(SE) + Context Retrieval (CR)**: The prompt was augmented with user-specific contextual information retrieved from prior tweets, using a similarity-based retrieval method similar to the one proposed in (Tan et al., 2024).

- **(SE) + (CR) + Fine-tuning (FT)**: The baseline model was fine-tuned on the full training set using the PEFT library (Mangrulkar et al., 2022).

For each of the 250 users and each of their 20 test tweets, we generated a candidate reply, totaling $5,000$ generated replies for each of the four configurations, for each model. The full prompt is reported in the Appendix .1.

## 4 Results

Our objective is to assess how effectively different LLM configurations can generate responses that are indistinguishable from human-authored content. In this section, we report (1) overall differences in stylistic and affective features between human and AI text, (2) model-level differences in stylistic fidelity, and (3) the impact of few-shot prompting, context retrieval, and fine-tuning on the realism of generated responses.

### 4.1 BERT-Based Turing Test Analysis

To evaluate each model configuration, we train a BERT-based classifier to distinguish between human- and AI-generated tweets, reporting two metrics: overall accuracy and accuracy on AI-generated text only. The ideal case is when the classifier performs at chance level (50% accuracy), indicating indistinguishability. As shown in Fig.1a, Google-gemma-3-4B-Instruct outperforms all other models by achieving lower classification accuracy, suggesting a greater ability to "fool" the classifier. Notably, achieving low accuracy is typically harder when focusing solely on AI-generated text.

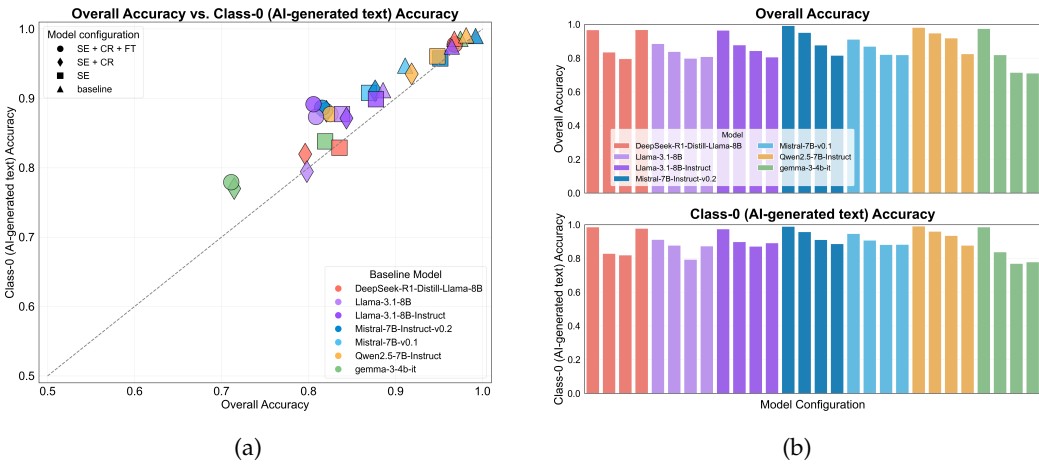

(a)                                        (b)

Figure 1: (a) Trade-off between overall classification accuracy and class-0 accuracy (i.e., accuracy restricted to AI-generated text). (b) Accuracies scores for different models, configurations, and metric. For the same model, configurations are ordered left to right: (BL), (SE), (SE) + (CR), (SE) + (CR) + (FT).

Additionally, performance varies within each model family depending on specific configuration choices, as shown in Fig. 1b: Adding stylistic examples (SE) and context (CT) consistently improves human-likeness, while the impact of fine-tuning (FT) is generally positive, with the except of Deepseek Model-R1-Distill-Llama 8B model.

### 4.2 Style and tone differences

We further examine which textual and stylistic features most influence the distinguishability of AI-generated content. We compare human- and AI-generated tweets across several metrics, including average word count, number of links and mentions, word length, punctuation, uppercase ratio, hashtag frequency, quotes, sentiment (via NLTK's SentimentIntensityAnalyzer (Hutto & Gilbert, 2014)), and toxicity (using the unitary/toxic-bert model (Hanu & team, 2024) based on the Detoxify approach (Hanu & the Unitary team, 2020)).

Results on the differences between the average values of these features computed among AI-generated tweets and among human-generate ones are shown in Fig. 2a. Notably, finetuned DeepSeek model exhibits excessive use of links, punctuation, and hashtags, correlating with increased average word length. This is likely the cause of its poor performances in the previous accuracy analysis. Conversely, AI-text generations from non-finetuned

Mistral-Instruct models exhibit high frequencies of hashtags, which are then corrected in
the finetuned model. More broadly, quotation marks, mentions, and hashtags are more
prevalent in AI-generated text in all model configurations (differences between the averages
in AI-generated text vs human-generated ones are consistently positive). Similarly, AI-
generated tweets tend to exhibit more positive sentiment and lower toxicity (with some
exceptions) than their human-written counterparts.

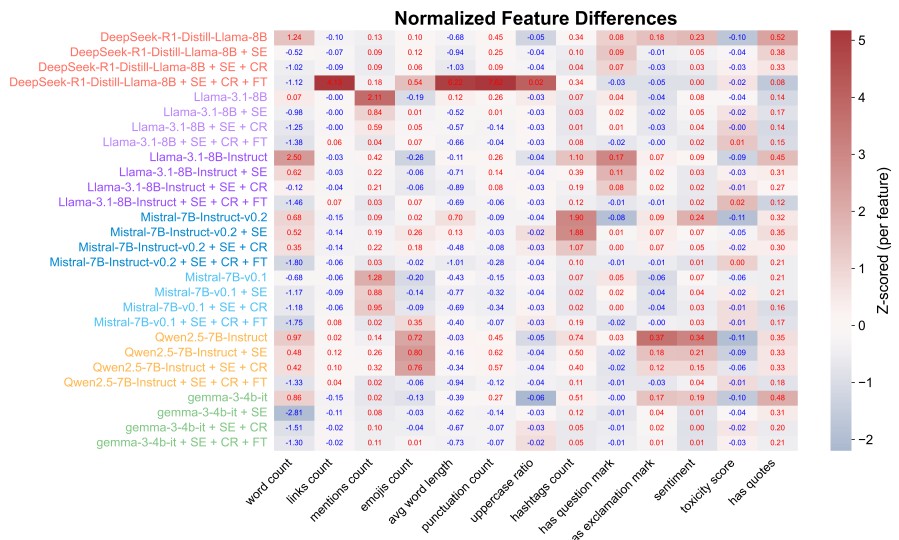

(a) Average differences per model configuration and feature. Numbers show the difference between
the average value among AI-generated text and the average value among human-generated one, with
positive values in red and negative values in blue; cell color indicates z-score (normalized per feature).

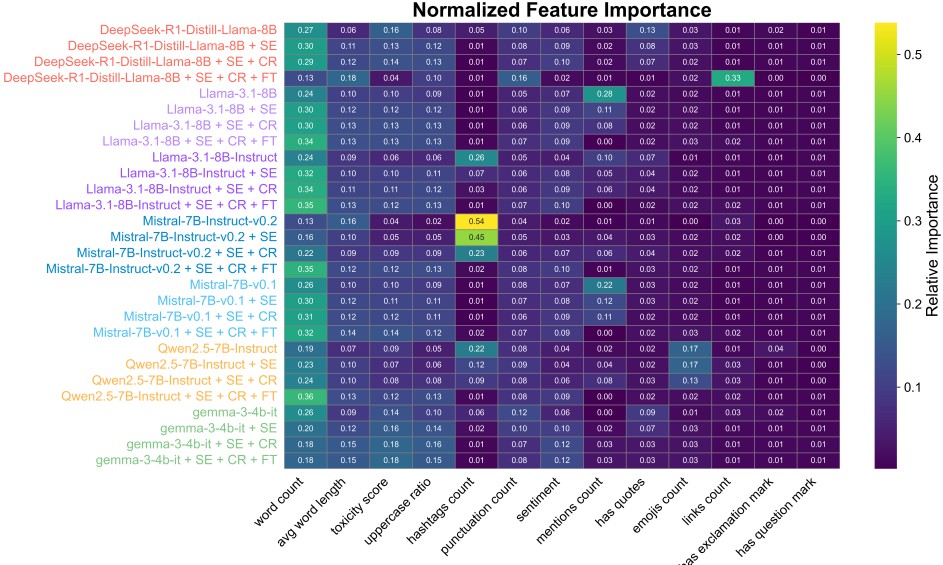

(b) Results of the analysis on the feature importance when using a random forest model to classify AI-
vs human-generated tweets. Columns are ordered by overall importance (across all the models), with
word count, average word length, and toxicity score being the most predictive features.

Figure 2

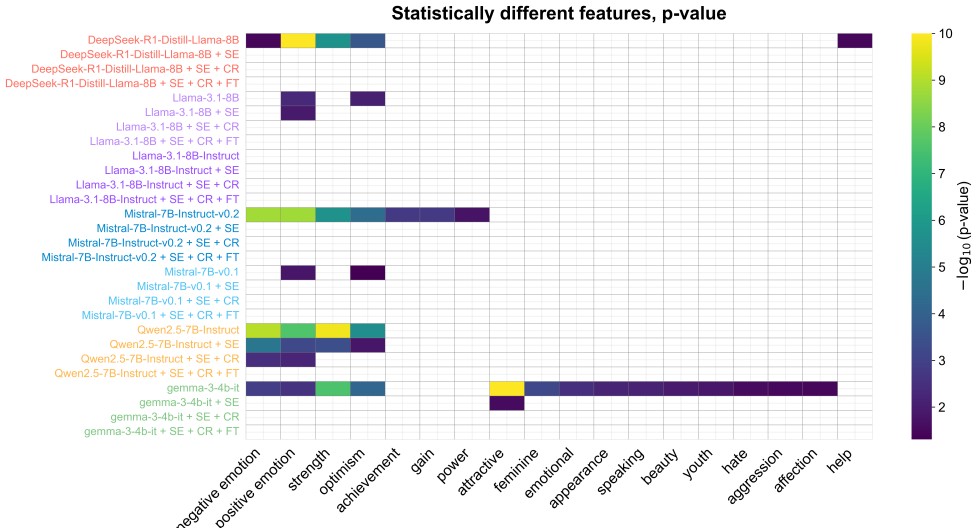

Figure 3: Results of the statistically significantly different features analysis through the Empath library. Columns are sorted according to decreasing average importance across the different model configurations.

### 4.3 Impact of stylistic features on BERT-predictions

To further investigate how specific stylistic attributes affect BERT's classification of AI-versus human-generated text, we trained a random forest classifier to perform the same task. We then analyzed feature importances across different model configurations. As shown in Fig. 2b, features such as word count, average word length, toxicity score, punctuation count, uppercase ratio, and sentiment consistently emerge as the most influential predictors. Some exceptions include: the finetuned DeepSeek model, where the number of links dominates, suggesting that this model tends to generate an unusually high number of links (as previously observed); the non-finetuned Mistral-Instruct and baseline Qwen models, where the number of hashtags is the most predictive feature; the non-finetuned Qwen model whose excessive usage of emojis makes AI-generated text easily detectable.

### 4.4 Empath analysis

To further understand the differences between AI- and human-generated text, we used the Empath library (Fast et al., 2016) and collected all the features that were measured to be statistically significantly different. According to the analysis reported in Fig. 3, baseline models are those that exhibit major differences, with negative and positive emotion, as well as strength and optimism being the most frequent features. Overall, baseline Gemma-3-4B-Instruct model is the one that exhibit the maximum number of different features.

## 5 ML-judge and Optimal selection

Given the high predictability of AI-generated tweets, we leveraged the feature importances identified in the classification task to improve the ability to fool the BERT classifier. To this end, we repeated the reply generation process, this time producing 20 candidate replies per tweet. This yielded a dataset of 250 users times 20 tweet prompts times 20 generated replies, totaling 100, 000 AI-generated responses per model-configuration. For each model-configuration, we then built a machine learning–based judge (ML-judge) to rank the generated replies from most to least likely to be misclassified as human, thereby identifying responses that are stylistically closer to authentic tweets. To prevent data leakage, we first removed duplicate replies (as some models frequently produced identical outputs)

and then partitioned the dataset into five user-based folds. For each fold, we trained
a random forest classifier on the remaining four folds and predicted, for each of the 20
candidate replies in the held-out fold, the probability of being classified as human.

The reply with the highest such probability was selected as the **optimal** response (in stylistic
terms), and later compared to the previous generation, sometimes referred as "random".
This procedure resulted in a dataset of $5,000$ optimal AI-generated replies, which we
combined with the $5,000$ human-generated replies from the original dataset. We then
applied the same analytical pipeline as in the previous step to compare the stylistic properties
of optimal and human responses.

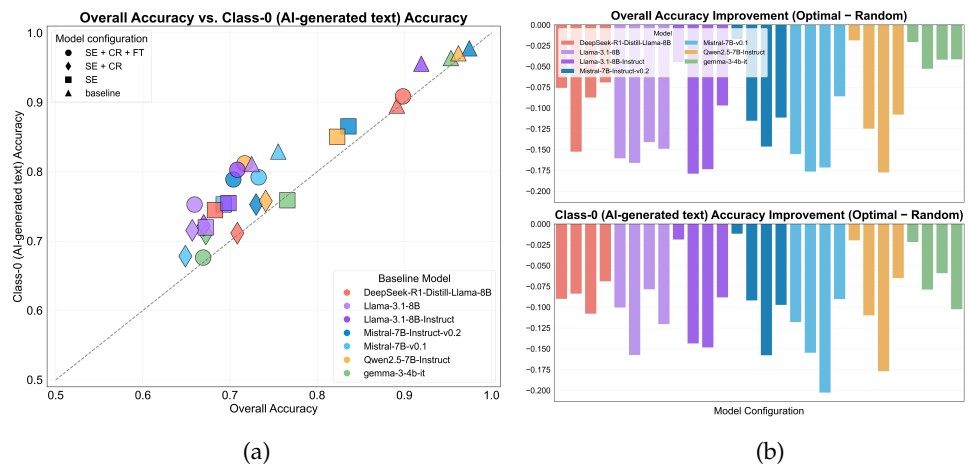

(a)          (b)

Figure 4: Accuracy results for optimal responses. (a) Trade-off between overall classification
accuracy and class-0 accuracy (i.e., accuracy restricted to AI-generated text). (b) Improve-
ment in the class-0 accuracy score for different models and configurations. For the same
model, configurations are ordered left to right: (BL), (SE), (SE) + (CR), (SE) + (CR) + (FT).

Results in Fig. 4 show that across all the model-configurations the use of an ML-judge to se-
lect an optimal response does improve in both accuracy metrics. In particular, even baseline
models are not fully detectable, and the majority of the more refined model configurations
have an overall accuracy of around 70% and a class-0 (i.e., AI-generated text restricted)
accuracy of around 75%. Notably, google-gemma3-4B-instruct model does not lead the
ranking anymore: Mistral-7B-v0.1 as well as Llama-3.1-8B do perform equally well, also
under different (mostly more refined) configurations.

Furthermore, we analyze the effect of ML-judge optimal selection by comparing the average
value of each feature in the optimal and random response approaches. As shown in Fig. 5a,
optimal responses tend to be longer (in word count) and include more links. Punctuation and
uppercase ratios also generally increase, with a few exceptions. Most model configurations
consistently reduce sentiment and increase toxicity, suggesting that, to make AI-generated
text less detectable, it is optimal to select responses that are slightly more toxic and less
positive. Finally, non-baseline Google-gemma models are less prone to sentiment reduction,
hinting at stronger guardrails toward generating positive content across all 20 candidates.

Lastly, Fig. 5b shows the importance of stylistic features in a random forest classification
task using the ML-judge's optimal responses. Compared to Fig. 2b, toxicity score is now
the most predictive feature for most models, suggesting the ML-judge fails to fully align
responses with human text. Average word length also ranks highly, along with uppercase
ratio, word count, and sentiment. Quote usage remains distinctive for many baseline models
(DeepSeek, Llama-Instruct, and Google-gemma), while hashtag and link frequency are the
top predictors for the Mistral baseline and DeepSeek fine-tuned models, respectively.

### 5.1 Empath analysis

Finally, we repeated the analysis through the Empath library for the optimal response scenario. Comparison between the results shown in Fig. 6 (in Appendix .2) as well as those previously reported in Fig. 3 indicate that there is no significant improvement, except for the fact that the difference in positive emotions is now more dominant (across different model configurations) than the difference in negative emotions.

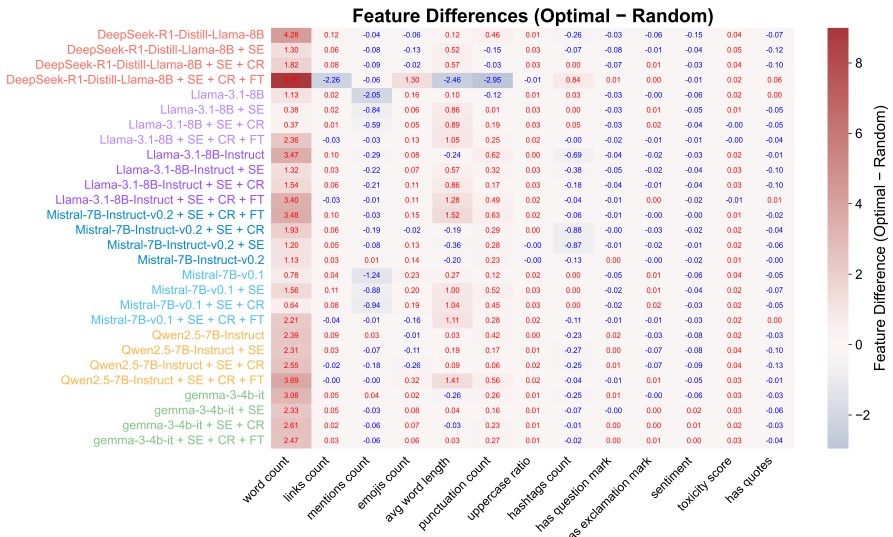

(a) Average difference between the **ML-judge optimal** response and the first implementation (random response) per model configuration and feature. Positive values in red and negative values in blue.

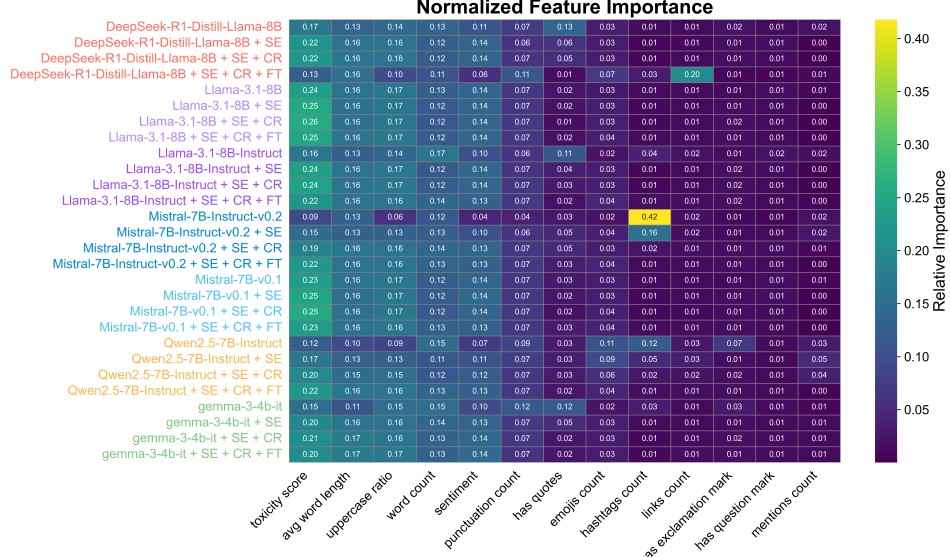

(b) Results of the analysis on the feature importance when using a Random Forest model to classify AI- vs human-generated tweets in the **ML-judge optimal** response scenario. Columns are ordered by decreasing overall importance (across all the models).

Figure 5

## 6   Discussion

Our findings offer a multifaceted view of how various LLM configurations perform when subjected to a BERT-based Turing test, as well as which linguistic and stylistic features are most instrumental in distinguishing AI- from human-generated content.

First, we observe that model architecture and configuration choices (e.g., fine-tuning, context injection, stylistic examples) significantly influence a model's ability to produce human-indistinguishable text. The Google-gemma-3-4B-Instruct model performs particularly well in the random response setting, consistently achieving the lowest classification accuracy by the BERT judge. However, once optimal responses are selected via an ML-based classifier (ML-judge), other models such as Mistral-7B and Llama-3.1-8B rise to the top.

Second, our feature-level analysis reveals persistent stylistic disparities between human and AI-generated text. Across nearly all models, AI-generated tweets contain more punctuation, links, and hashtags, and exhibit higher positivity and lower toxicity. These stylistic patterns tend to be amplified in certain configurations (e.g., DeepSeek fine-tuned models), rendering them easier for BERT to detect. Even after optimization with the ML-judge, complete alignment with human style is not achieved. While some features (e.g., average word count, punctuation usage) get closer to human baselines in optimal responses, others (e.g., sentiment and toxicity) do not exhibit significant improvements.

Importantly, our feature importance analysis shows that attributes like toxicity, sentiment, and formatting (uppercase ratio, word length, punctuation) dominate classifier decisions, both in the random and optimal response settings. The shift in dominant features from random to optimal responses (e.g., increase in toxicity's predictive power) indicates that ML-based selection does not equally neutralize the key indicators of artificiality.

Lastly, Empath analyses confirm that despite some emotional refinement by the ML-judge, major gaps in affective and psychological markers remain, highlighting the challenge of true semantic and affective alignment.

## 7   Conclusion

This study has provided a systematic and detailed assessment of the stylistic indistinguisha-bility of LLM-generated text from human-authored tweets, using a BERT-based classifier as a Turing test proxy. Our results demonstrate that while certain LLMs – particularly when enhanced through stylistic examples, context retrieval, and fine-tuning – can reduce their detectability, no configuration fully escapes classification. Even with the aid of a machine learning–based judge for selecting optimal outputs, LLM-generated text retains detectable stylistic and affective signatures that set it apart from genuine human discourse. In particular, AI-generated replies consistently exhibit higher positivity, lower toxicity, and subtle divergences in structure, sentiment, and formatting.

These findings point to two critical insights. First, achieving human-like generative text goes beyond increasing model size or architectural sophistication. It requires fine-grained conditioning and intelligent output selection that capture deeper psychological and affective patterns, not just surface-level style. Second, even sophisticated selection mechanisms, such as our ML-judge, are insufficient to eliminate persistent signals of artificiality.

This has important implications for researchers using LLMs in simulations and experimental studies. Our analysis suggests that current models often fall short of producing text that is fully realistic in tone and style, raising concerns about the validity of social simulations based on generative agents. Prompt-based calibration alone is unlikely to achieve the necessary stylistic fidelity. Instead, progress will depend on more advanced methods that combine architectural innovations, nuanced control over affective dimensions, and robust output selection. Addressing these challenges is essential for building generative agents that can serve as credible proxies for human behavior in computational social science.

## Acknowledgments

This research was supported by the Swiss National Science Foundation (SNSF) under grant number [IZTAZ1_223462], as part of the DemDialogueAI project funded through the Trans-Atlantic Platform's Democracy, Governance, and Trust initiative. For more information, see https://www.transatlanticplatform.com/demdialogueai/.

## Ethics Statement

This study builds on social media data originally collected by Cerina (2025) via the official Twitter/X API. The dataset consists exclusively of publicly available posts and associated metadata that do not include personal or sensitive information. In accordance with institutional guidelines and applicable legal frameworks, including data protection regulations, this research does not constitute human subjects research and therefore did not require ethical review. Nonetheless, we took care to handle the data in ways that respect user privacy and minimize potential risks of harm.

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

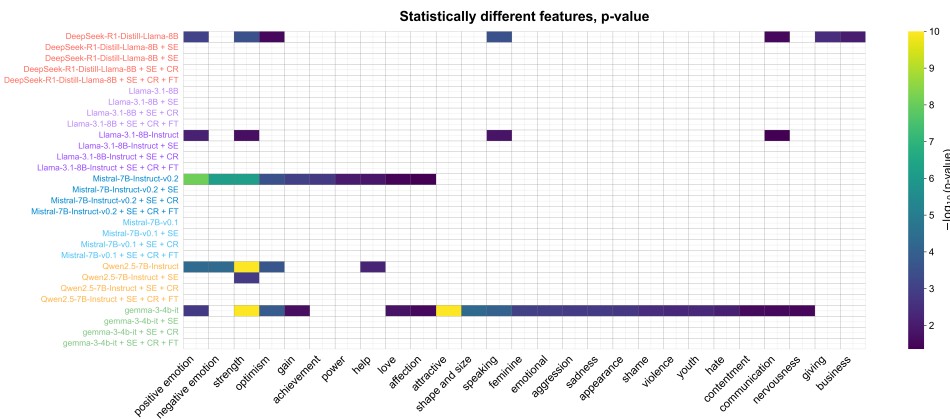

Figure 6: Analysis of the statistically significantly different features results of the Empath library analysis for the **optimal response.**

# Appendix

## .1 Prompt

The full input to the LLMs was a structured prompt designed to simulate a realistic conversational setup.

```
def build_prompt(username, persona_examples, conversation_history, retrieved_context=""):
    prompt = "[Instruction] You are @{username}. Continue the conversation naturally
        adding a concise (one sentence) tweet reply.\n"
    if persona_examples:
        examples = "\n".join(f"- {ex}" for ex in persona_examples)
        prompt += f"[Writing Style] These are some tweets that represent how
        @{username} writes:\n{examples}\n\n"
    if retrieved_context:
        prompt += f"[User Retrieved Context] This is some useful context retrieved
        from @{username}'s history \n" + retrieved_context + "\n\n"
    if conversation_history:
        prompt += "[Conversation] " + "\n".join(conversation_history) + f"\n{username}:"
     return prompt
```

## .2 Empath analysis

For completeness, in Fig. 6 we report the results of the Empath analysis in the optimal response scenario. Similarly to the random response scenario, Google-gemma-3-4B-Instruct baseline model is the one with the highest number of statistically significant different features, followed by Mistral-7B-Instruct-v0.2 baseline. Positive and negative emotion, as well as strength and optimism are the features that are more frequently found to be different.

