# OpenReview forum: "Can LLMs Imitate Social Media Dialogue? Techniques for calibration and BERT-based Turing-Test"
_colmweb.org/COLM/2025/Workshop/Social_Sim — Social Sim'25_

### Official Review · Reviewer_Y3Ex · 2025-07-13
**Promising Research Question, but Experimental Design Needs Improvement**

**Rating:** 6
**Overall Assessment:** 3
**Confidence:** 4

**Review:**

The research question is both fundamental and highly relevant to Social Sim'25, addressing the simulation of human likeness in online environments. By focusing on Twitter/X texts as the experimental dataset, the paper grounds its analysis in a realistic online platform context. The experiments are of generally good quality, benchmarking multiple open-weight LLMs and attributes, and the writing is clear and well-structured, contributing to overall clarity.

However, several weaknesses limit the paper’s overall impact. The experimental design does not validate the reliability of the ML model used as a judge, raising concerns about the soundness of the evaluation. The exclusive reliance on Twitter/X data further restricts the generalizability of the results. While the central idea of simulating digital human-likeness is novel and promising, the selected stylistic and affective features—such as word count and punctuation usage—are too shallow to capture the complexity of human language. Moreover, poorly designed visualizations, especially in Figures 1a and 4a, hinder interpretability and diminish the clarity of key findings.

**Comments Suggestions And Typos:**

* Validate the judge ML model by comparing its judgments with human annotations, to ensure reliability.
* Extend the dataset to include texts from other online platforms.
* Include a clearer definition of stylistic and affective features and add deeper stylistic and affective features (e.g., sentiment, tone, lexical variety) for a more meaningful analysis.
* Improve visualizations to make it clearer.
* Generalize to other tasks to have a broader discussion on this topic.
* Experiment with top-performing LLMs.

**Paper Summary:**

This paper answers the question *To what extent can LLMs effectively simulate human writing in terms of style and communicative effectiveness within online environments?*

To answer this research question, the authors use various prompt types to generate LLM outputs and train a machine learning model to distinguish them from human-written texts. They further analyze ML model parameters to identify which textual features contribute most to the prediction.

This work contains three main contributions: it proposes a novel human-likeness evaluation framework beyond traditional human judgment, systematically benchmarks open-weight LLMs under diverse settings, and analyzes stylistic differences between human and machine-generated texts.

**Relevance:**

5

**Summary Of Strengths:**

* The research question is important and very relevant to the topic covered by Social Sim'25.
* It benchmarks a diverse set of open-weight LLMs and multiple prompt formats, increasing the generalizability of its insights.
* The study design is generally well-structured, and the paper is clearly written and easy to follow.
* The proposed evaluation framework is reproducible and could be applied to other human-likeness assessments in future work.
* The work offers a novel perspective by situating LLM evaluation in online environments, which is an underexplored but increasingly important domain.

**Summary Of Weaknesses:**

* The paper lacks validation of the machine learning model used as a judge, raising concerns about the reliability of the evaluation results.
* The dataset is restricted to Twitter texts, which limits the generalizability and scope of the findings across online environment.
* Some visualizations are not clear and hard to interpret.
* The features chosen to represent stylistic and affective features (e.g., word count, mention count, punctuation count, uppercase ratio) are superficial. They fail to reflect deeper stylistic traits (e.g., formality, tone, sentence structure) or emotional dimensions (e.g., sentiment polarity, emotional intensity), weakening the central claims about human-likeness in LLM output.
* The experiments rely mostly on smaller models (e.g., 8B), which limits the strength of the benchmarking results.

---

### Official Review · Reviewer_rFdo · 2025-07-16

**Rating:** 4
**Overall Assessment:** 2
**Confidence:** 4

**Review:**

The paper argues (in section 1) that model-generated responses should be "believable" to enhance simulations. However, they only analyze statistical and stylistic features such as word count, punctuation, etc, and do not analyze the content. Without accounting for the content, their claims about how this method can enable mitigating bias are unfounded, as the bias propagated by text is mostly in the content and not the stylistic features. Furthermore, the studied features are rudimentary and lower-level. Analyzing the higher-order features, such as sentence structure (POS/Argument structure, etc), would be more beneficial.

1. Since the study mentions that toxicity and emotions differ between the generated and actual data (in Section 6), a detailed description of the dataset could help. How toxic and emotionally laden were the original replies?
2. Since the temperature was set to 0.8 (Line 134), to ensure generalizability, the implemented method should generate multiple responses for the same seed prompt and observe the distribution of the linguistic features, instead of analyzing only 1 response for a prompt.
3. Details about fine-tuning the models are missing (Line 144)
4. Line 156 mentions that a BERT-based classifier was trained to distinguish human vs AI-generated text. However, details of the classifier are missing. The details are crucial since the analysis hinges on the reliability and performance of the classifier.
5. Line 209 mentions training an ML-judge to rank the generated replies. Is this judge the same as the random forest classifier? Also, crucial details about the ML-judge pertaining to its performance are missing.

**Comments Suggestions And Typos:**

1. Line 125-126: You have already defined LLM. No need to define it again.
2. The Acknowledgement section reveals the identity of the author

**Paper Summary:**

The paper analyzes how human-generated content differs from AI-generated text. They derive several linguistic features from text and use a BERT-based classifier to compare them. They introduce an ML-judge to rank the synthetically-generated responses to only choose the best responses, and observe gains in the ability to "fool" the BERT-based text discriminator.

**Relevance:**

3

**Summary Of Strengths:**

1. The paper is easy to follow.
2. The research question of how original and AI-generated text differ is interesting.

**Summary Of Weaknesses:**

Mentioned in the review

---

### Meta-Review · Area_Chair_jc5s · 2025-07-21

**Recommendation:** Accept

**Metareview:**

Please incorporate the feedback from the reviews.